# AIDmut-Seq: a Three-Step Method for Detecting Protein-DNA Binding Specificity

Feixuan Li,[a] Xiao-Yu Liu,[c] Lei Ni,[b] Fan Jin[b]

[a]Hefei National Research Center for Physical Sciences at the Microscale, Department of Polymer Science and Engineering, University of Science and Technology of China, Hefei, People's Republic of China

[b]CAS Key Laboratory of Quantitative Engineering Biology, Shenzhen Institute of Synthetic Biology, Shenzhen Institutes of Advanced Technology, Chinese Academy of Sciences, Shenzhen, People's Republic of China

[c]School of Medicine, Southern University of Science and Technology, Shenzhen, People's Republic of China

**ABSTRACT** Transcriptional factors (TFs) and their regulons make up the gene regulatory networks. Here, we developed a method based on TF-directed activation-induced cytidine deaminase (AID) mutagenesis in combination with genome sequencing, called AIDmut-Seq, to detect TF targets on the genome. AIDmut-Seq involves only three simple steps, including the expression of the AID-TF fusion protein, whole-genome sequencing, and single nucleotide polymorphism (SNP) profiling, making it easy for junior and interdisciplinary researchers to use. Using AIDmut-Seq for the major quorum sensing regulator LasR in *Pseudomonas aeruginosa*, we confirmed that a few TF-guided C-T (or G-A) conversions occurred near their binding boxes on the genome, and a number of previously characterized and uncharacterized LasR-binding sites were detected. Further verification of AIDmut-Seq using various transcriptional regulators demonstrated its high efficiency for most transcriptional activators (FleQ, ErdR, GacA, ExsA). We confirmed the binding of LasR, FleQ, and ErdR to 100%, 50%, and 86% of their newly identified promoters by using in vitro protein-DNA binding assay. And real-time RT-PCR data validated the intracellular activity of these TFs to regulate the transcription of those newly found target promoters. However, AIDmut-Seq exhibited low efficiency for some small transcriptional repressors such as RsaL and AmrZ, with possible reasons involving fusion-induced TF dysfunction as well as low transcription rates of target promoters. Although there are false-positive and false-negative results in the AIDmut-Seq data, preliminary results have demonstrated the value of AIDmut-Seq to act as a complementary tool for existing methods.

**IMPORTANCE** Protein-DNA interactions (PDI) play a central role in gene regulatory networks (GRNs). However, current techniques for studying genome-wide PDI usually involve complex experimental procedures, which prevent their broad use by scientific researchers. In this study, we provide a *in vivo* method called AIDmut-Seq. AIDmut-Seq involves only three simple steps that are easy to operate for researchers with basic skills in molecular biology. The efficiency of AIDmut-Seq was tested and confirmed using multiple transcription factors in *Pseudomonas aeruginosa*. Although there are still some defects regarding false-positive and false-negative results, AIDmut-Seq will be a good choice in the early stage of PDI study.

**KEYWORDS** activation induced cytidine deaminase, *Pseudomonas aeruginosa*, protein-DNA interactions

Address correspondence to Lei Ni, lei.ni@siat.ac.cn, or Fan Jin, fan.jin@siat.ac.cn.

The authors declare no conflict of interest.

The identification of target promoters for transcription factors (TFs) remains to be a challenge when establishing genetic regulatory networks. Current methods for detecting protein-DNA binding specificity rely mainly on the sequencing of TF-bound enriched DNA fragments, such as chromatin immunoprecipitation by sequencing (ChIP-

seq), systematic evolution of ligands by exponential enrichment (SELEX), and DNA affinity purification sequencing (DAP-seq) (1–4). In ChIP-seq, genomic DNA was cross-linked to TFs *in vivo* and fragmented into small fragments. Those TF-bound DNA fragments were enriched using TF-specific antibody. In SELEX, a library of randomized 14-bp DNA ligands was constructed and incubated with interested TFs which are immobilized in a 96-well plate. Unbound DNA ligands were washed and eluted, the remaining TF-bound DNA ligands were amplified and subjected to sequencing. In DAP-seq, genomic DNA was sheared into 200-bp fragments and ligated with sequencing adaptors to generate a DNA library. Interested TFs were expressed *in vitro* with a fused tag and immobilized on magnetic beads. The TF-bounded DNA fragments were enriched via incubation, washing and elution similar to the SELEX method. The above three methods often involve techniques that are challenging for many researchers, like DNA fragmentation, end repair, adaptor ligation, and affinity purification of TFs. Combining these techniques into a workflow further complicates experiments, which could lead to failure in some cases.

Besides, SELEX and DAP-seq are *in vitro* methods and may fail to detect TF-binding sites in cases where intracellular cofactors are involved in TF activation. For example, the activities of two-component response regulators are dependent on their phosphorylation by cognate sensors, and some TFs are activated by small molecules such as quorum sensing signals and metabolic intermediates. In addition, *in vitro* methods are defective when the activation condition for a TF is unknown. Under these conditions, *in vivo* methods have an advantage over *in vitro* methods for studying protein-DNA interactions.

DNA adenine methyltransferase identification and sequencing (DamID-seq) and Calling cards represent two *in vivo* strategies to search for TF-binding sites across the genome. They act by introducing base methylation or transposon insertion near TF-binding sites, followed by sequencing of enriched sequences flanking the genome editing sites (5, 6). In DamID-seq, an *E. coli* adenine methyltransferase was fused to interested TFs, which can methylate adenine base in GATC motifs near the sites of TF-DNA interactions when expressed *in vivo*. The resulted genomic DNA was digested with DpnI that only cut adenine-methylated GATCs, then the DNA fragments were ligated with adaptors, further digested with DpnII (cut all GATC motifs) into smaller fragments. These DNA fragments were ligated with a second adaptor so that only the fragments containing DpnI-digested ends have adaptors on both sides, thus can be amplified for sequencing. In the Calling cards method, interested TFs were fused with an integrase, which can initiate Ty5 integration into the genome near the TF binding sites. Genomic DNA were extracted from cells undergone Ty5 transposition, and then digested with Ty5-specific restriction enzyme that cut at sites away from their recognition motif. In this way, part of the genomic sequences flanking the TF-binding sites were included in the digested small fragments. These fragments were self-ligated to favor recircularization, amplified using inverse PCR, and sequenced. As can be seen from the above instructions, DamID-seq and Calling cards still involve multiple steps of DNA digestion, ligation, and amplification.

Recently, another *in vivo* method named 3D-seq was developed by fusing the cytidine deaminase DddA to the 3′ end of TFs (7). TF-binding sites were directly mapped by searching C-to-T or G-to-A transition peaks on the genome. 3D-seq simplified the experimental procedure, while still involving in-frame fusion of *dddA* with TF genes on the genome. Besides, DddA targets double-stranded DNA and initiates multiple single nucleotide polymorphisms (SNPs) within a broad range of 10 kb, which is lethal to bacterial cells when essential genes were inactivated by mutation. Therefore, 3D-seq requires the exogenous expression of a DddA cognate immunity determinant induced by arabinose to antagonize DddA's activity, so that genomic mutations only occurs when arabinose was removed. The 10 kb SNP window also precludes 3D-seq in distinguishing close TF-targets on the genome. For example, the bis-(3′–5′)-cyclic dimeric GMP (c-di-GMP) dependent transcriptional regulator FleQ in *Pseudomonas aeruginosa* has three binding sites that are less than 200 bp apart on the *cdrA* promoter, while they exhibited only one summit in the 3D-seq map (7, 8). In

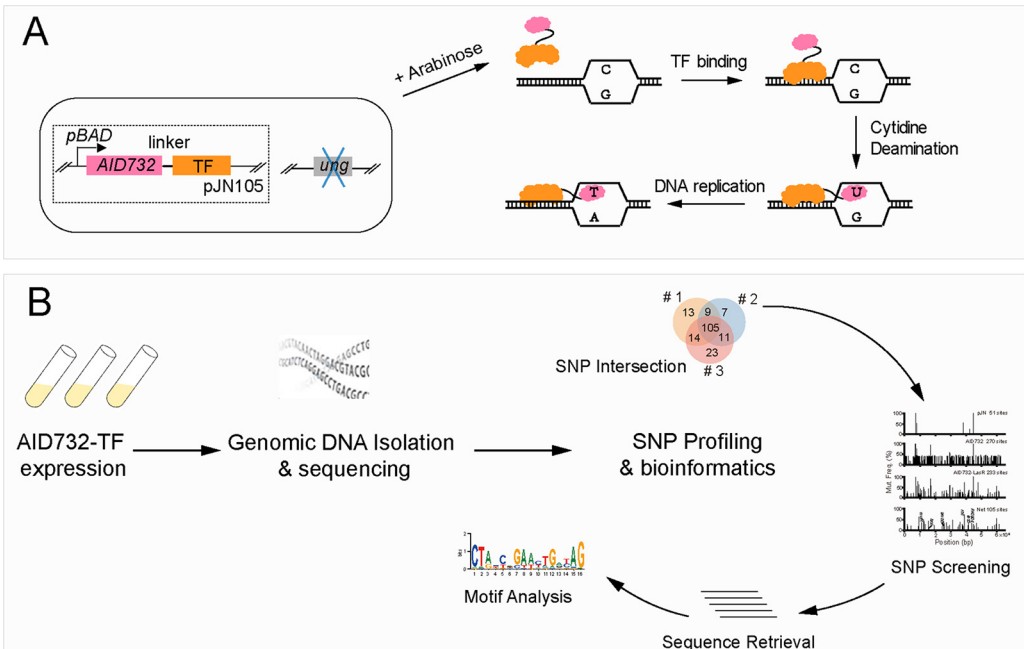

**FIG 1** Overview of the AIDmut-Seq method. (A) Schematic diagram of AIDmut-Seq. *ung*, the genomic uracil-DNA glycosylase gene was deleted. The *AID732-TF* fusion gene was expressed under an arabinose-responsible promoter *pBAD* in the pJN105 plasmid. AID732-TF binds TF-binding boxes on the genome and induces cytosine base deamination near the binding sites. (B) Workflow of the AIDmut-Seq method involving three major steps: AID732-TF fusion protein expression, genomic DNA isolation and whole-genome sequencing, and single nucleotide polymorphisms (SNPs) profiling. Three parallel experiments were conducted to measure the genomic binding sites of each TF. Common SNPs obtained in three experiments were screened first to eliminate background SNPs, and then submitted to a bioinformatic tool for further sequence retrieval, annotation, and motif analysis.

this study, we aimed to develop a simple method that is achievable by junior researchers and research groups with little experimental experience in molecular biology.

The activation induced cytidine deaminase (AID) from mouse can deaminate C to U in targeted DNA sequences when accessing single-stranded DNA, resulting in C-T or G-A substitutions following DNA replication (9–11). Compared to DddA whose deaminase activity does not depend on the accession of single-stranded DNA, AID should produce fewer genomic mutations, and therefore be less harmful to cells. And it is relatively small with a molecular weight of 24 kDa, facilitating its fusion with other proteins to achieve directed mutation. As exemplified by the dCas9-guided MS2-AID mutation generator, in which AID was fused to a hairpin-binding protein and was recruited by dCas9 to generate mutations at target genomic sites (12). We hypothesized that TF-fused AID could introduce mutations near TF-binding sites, thus generating SNPs around these sites (Fig. 1A), which can be directly detected through whole-genome sequencing. Here, we reported our validation and optimization of this AID-introduced mutation approach combined with sequencing, referred to as AIDmut-Seq, in the model pathogen *P. aeruginosa*. AIDmut-Seq generates several discrete SNPs that are closely near the TF-targets, thus has a higher resolution compared to current methods. AIDmut-Seq is performed *in vivo*, is time-saving, and is easy to operate. The whole workflow for AIDmut-Seq consists of only three steps (Fig. 1B), takes 6 days before sequencing, and could cost approximately $130 for each TF.

## RESULTS

**Establishment of the AIDmut-Seq platform.** To test the method, we started by fusing AIDΔ to the 5' end of the major quorum sensing regulator LasR with a flexible linker (G$_8$) in *P. aeruginosa* strain PAO1. AIDΔ is a variant of the mouse cytidine deaminase AID, with its nuclear export signal deleted and has three amino acid substitutions (K10E, E156G, T82I) (12, 13). The DNA fragment was cloned into the broad-host vector pJN105, in

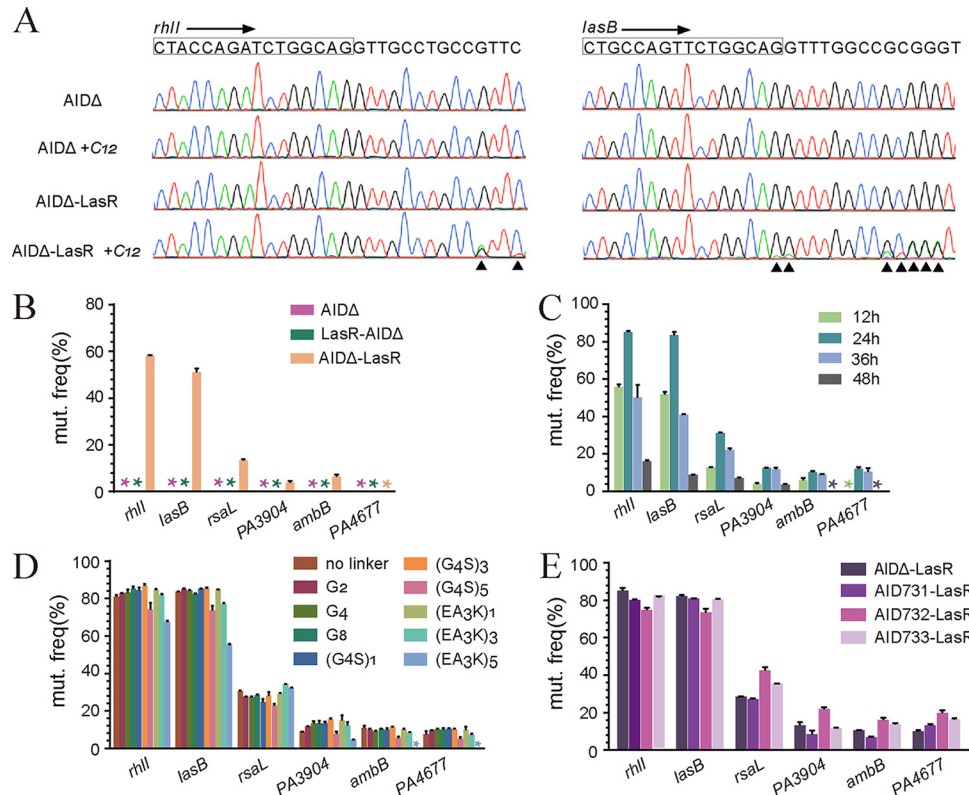

**FIG 2** AIDmut-Seq application to the major quorum sensing regulator LasR in *P. aeruginosa* and method optimization. (A) AIDΔ fused to the 5′ end of LasR initiated G-to-A or C-to-T mutations nearby the LasR-binding box of *rhlI*, *lasB* promoter in the PAO1 genome. Black triangles indicate the mutation positions. (B) Mutation initiation activities by AIDΔ fused to the 5′ or 3′ end of LasR in six LasR-responsive promoters. (C) Mutation initiation activities by AIDΔ-LasR in different arabinose induction periods. (D) Mutation initiation activities by AIDΔ-LasR using six flexible and three rigid linkers between them, with 24-h arabinose induction. (E) Mutation initiation activities by different AID variants fused to the 5′ end of LasR using G8 linker, with 24-h arabinose induction.

which the expression of AIDΔ-LasR is controlled by an arabinose-inducible promoter. The *ung* gene, which encodes an uracil-N-glycosylase involved in base excision repair, was deleted to eliminate the repair of AIDΔ-induced mutations. We also deleted the *lasI* and *rhlI* genes, which are responsible for autoinducer synthesis, to facilitate the manipulation of LasR activity via externally added N-3-oxo-dodecanoyl-homoserine lactone (3OC12-HSL) inducer (14). The *lasR* gene in PAO1 genome was also deleted to prevent competitive binding of native LasR and AIDΔ-LasR with DNA boxes. After 12 h of incubation with 20 μM 3OC12-HSL and 0.4% arabinose inducers, several G-A or C-T mutations were detected near the previously identified LasR-binding boxes (LBB) in the *rhlI* and *lasB* promoters (Fig. 2A). No mutations were observed in these promoters when the inducers were not added, or when AIDΔ-LasR was replaced by AIDΔ (Fig. 2A), indicating that AIDΔ-LasR was active in both specific DNA binding and cytidine deamination.

We next investigated the effects of using different fusion directions, linker types, AID variants, and induction time on the activity of LasR-fused AID in generating specific mutations. Six LasR-responsive promoters with different LBBs were tested (15). We used the frequency of G-A or C-T conversion as the criterion to determine the activity of fused AID. In case of multiple mutation sites within one promoter, the site with the highest frequency of mutation was chosen to represent the degree of mutation for that promoter. AIDΔ-LasR produced high mutation frequencies in the *rhlI* and *lasB* promoters, whereas a lower frequency of mutation was observed in the *PA3904* and *ambB* promoters, and no mutation was observed in the *PA4677* promoter (Fig. 2B). The distinct ability of AIDΔ-LasR to produce mutation in different promoters may arise from

the different binding affinities of their DNA binding box to LasR, as well as the different transcriptional activities of these promoters, given that the activities of AID-family deaminases are dependent on their access to single-stranded DNA (16).

When AIDΔ was fused to the 3′ end of LasR, no mutations were detected in any of the promoters tested (Fig. 2B), probably due to a loss-of-function effect in the DNA-binding domain of LasR. Thus, we fused AIDΔ to the 5′ end of LasR in subsequent experiments. Besides, mutations generated in the promoter or coding sequences of a gene can affect its expression level or result in a loss of gene function, both of which may reduce the growth rate of a cell, leading to a decreased proportion of the mutated cells in the whole population after multiple generations. This will reduce the detected mutation frequencies of some mutations. To minimize the impact of possible growth rate reduction on the detection of genomic mutations, we added the inducers at an initial $OD_{600}$ of 1.0 and diluted bacterial culture 5× for each 12 h of shaking. Using this approach, the bacteria can take several generations to complete the C-T conversions induced by AIDΔ, while the population will not experience too many generations, which will eliminate those mutants with low growth rates. We observed the highest mutation frequency in all promoters with a culture time of 24 h (two rounds of 12-h culture). Prolonged induction with arabinose and 3OC12-HSL resulted in decreased frequency of mutations (Fig. 2C). In addition, nine different linkers, including six flexible linkers and three rigid linkers, were tested. Except for two long linkers, $(G_4S)_5$ and $(EA_3K)_5$, that led to a significant reduction in mutation frequency, other linkers exhibited similar effects on the activity of AIDΔ-LasR (Fig. 2D). We also tested AID variants with higher deaminase activities (13). All tested AID variants showed similar extents of mutation generation in the *rhll* and *lasB* promoters (Fig. 2E). However, AID732 exhibited an advantage in producing mutations in other nonsensitive promoters and thus improved the detection limit for the fusion system (Fig. 2E).

The expression level of AID732-TF fusion protein is a key factor that affects the output of AIDmut-Seq. Too low expression level of the fusion protein will result in failure in detecting expected binding sites, while too high expression level of AID732-TF may lead to the detection of a large number of binding sites with weak binding affinity. Suppose TF-DNA binding belonging to the first-order hill equation, AID732-TF expression at a level of $[TF] = k_d$ will result in a 50% probability of DNA being bound by TF. Here, $k_d$ is the equilibrium dissociation constant of TF-DNA binding. For common promoters, $k_d$ is generally within the range of 1 nM to 1 $\mu$M. We quantified the expression level of our arabinose-inducible system (Supplemental File 1: Fig. S1) and expressed AID732-TF at an intracellular concentration about 7 to 8 $\mu$M.

In summary, we chose to fuse AID732 to the 5′ end of LasR with $G_8$ linker and incubate bacteria with 0.4% arabinose for 24 h before sequencing.

**Identification of AID732-LasR-initiated mutations via whole-genome sequencing.** First, whole-genome sequencing of the background Δ*lasl*Δ*rhll*Δ*ung*Δ*lasR* strains carrying pJN105 void vector (represented as pJN_b1) and carrying AID732-pJN105 vector (represented as AID732pJN_b1) was conducted. We found 51 SNPs in pJN_b1 relative to the PAO1 wild-type strain (Fig. 3A, Supplemental File 2), arising from stochastic mutations during bacterial culture. Two hundred and seventy SNPs were found in the genome of AID732pJN_b1 (Fig. 3A, Supplemental File 2), reflecting the biased AID732 mutation sites in PAO1 genome. The SNPs detected in pJN_b1 and AID732pJN_b1 were excluded when they appeared in the sequencing results of Δ*lasl*Δ*rhll*Δ*ung*Δ*lasR* strain carrying AID732-LasR-pJN105 (represented as AID732LasR_b1). There was a high-frequency mutation region in the Pf4 prophage site (genes from *PA0715* to *PA0729*) of the PAO1 genome, for reasons which remain unclear. SNPs detected in this region were also excluded. Second, sequencing depths of 100×, 200×, and 500× were tested for an identical AID732LasR_b1 sample. Greater sequencing depths detected slightly more SNPs, but sequencing performed at 100× was determined to be enough to detect most SNPs (Fig. 3B).

Stochastic deamination by AID732-LasR, which is independent of specific LasR-DNA binding, will produce false-positive SNPs in bacterial genome. To eliminate these false-

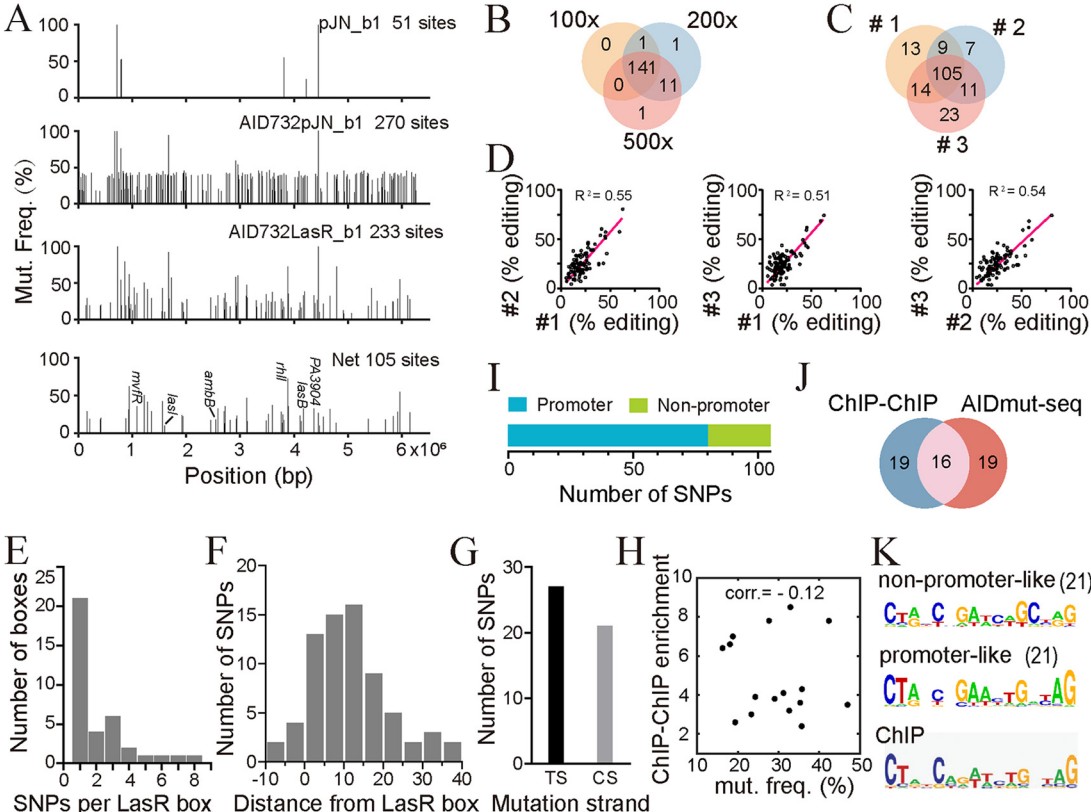

**FIG 3** Analysis of the whole-genome sequencing data of AIDmut-Seq for LasR. (A) SNPs detected in the background of pJN_b1 and AID732pJN_b1 strains were excluded from the 233 detected SNPs in the AID732LasR_b1 strain to acquire 105 LasR-specific SNPs. (B) Venn diagram for three net-SNP genome sequencing results with sequencing depth of 100×, 200×, and 500× for an identical AID732LasR_b1 sample. (C) Venn diagram for the net-SNP genome sequencing results of three independent AID732LasR_b1 samples. (D) Mutation frequency reproducibility of common SNPs detected in three independent experiments for the AID732LasR_b1 strain. $R^2$ represents the coefficient of determination for linear regression. (E) Statistics of the number of SNPs generated near LasR-binding boxes. (F) Statistics of the distance (measured according to the distance of a specific SNP to the nearest base in the LasR-binding box) of SNPs from their corresponding LasR-binding boxes. Negative distance means that SNPs appeared in the box. (G) Statistics of the number of C-to-T conversions detected at the template strand (TS) or the coding strand (CS). (H) Scatterplot of ChIP-ChIP enrichment score against the corresponding mutation frequency measured in AIDmut-Seq. For promoters with multiple SNPs, the biggest mutation frequency was used. corr., Correlation coefficient. (I) Distribution of the 105 net SNPs detected in the genome of the AID732LasR_b1 strain. (J) Venn diagram for LasR-responsive promoters detected by AIDmut-Seq and promoters detected by ChIP-chip. (K) DNA motifs generated from the flanking 200 bp sequences of SNPs belonging to promoter-like or nonpromoter-like regions, compared with a former established PWM model. DNA motifs were calculated using the online MEME tool. The number in parentheses after motifs indicates the number of promoters or genes used to calculate the motif.

positive SNPs, we extracted the shared mutations of the AID732LasR_b1 genome in three independent experiments. According to our sequencing results of AID732pJN_b1, the average probability of stochastic mutation ($p_0$) generated by AID732 through the experimental procedure was less than $10^{-4}$ per base pair. We assume that AID732-LasR has a similar probability to generate stochastic mutation. Thus, the probability of one mutated base pair that occurs in both three independent experiments is $p_0^3$. Then the average number of shared mutations from three independent experiments was $N \cdot p_0^3$. Here, $N$ is the total number of base pairs of the genome which is within the range of $10^6 \sim 10^7$ for common bacterial species. Therefore, $N \cdot p_0^3$ is far less than 1.0. That is, less than one shared stochastic mutation can be detected from three independent experiments. Thus, those stochastic false-positive results can be eliminated.

In total, 105 SNPs were extracted for AID732LasR_b1, accounting for 74%, 80%, and 69% of all SNPs detected in the three parallel samples (Fig. 3C). These data indicate that the repeatability of mutation detection is high, and the level of nonspecific mutations generated by AID732-LasR was relatively low. The $R^2$ for the mutation frequencies

of the parallel experiments were 0.55, 0.51, and 0.54 (Fig. 3D), representing a moderate repeatability in the measurement of mutation frequency.

**Characterization of AID732-LasR-initiated mutations and validation of LasR binding.** We next analyzed several characteristics of AID732-LasR-initiated mutations. We found that (i) only one SNP was observed near most LasR-binding sites in the genome (Fig. 3E); (ii) the vast majority of SNPs occurred within 40 bp from LBB (Fig. 3F); (iii) C-T mutations did not exhibit significant bias toward the template strand or the coding strand (Fig. 3G); (iv) The measured mutation frequencies of SNPs on target promoters are not correlated with the corresponding enrichment score from ChIP-ChIP experiment (15) (Fig. 3H).

Of the 105 SNPs identified in the genome of the AID732LasR_b1 strain, 80 were found in the sequence of 35 promoter-like (intergenic, upstream of at least one gene) regions, including 16 previously reported LasR-regulated promoters (Fig. 3I and J). The remaining 25 SNPs were located in either the coding sequences or nonpromoter-like intergenic regions (Supplemental File 3). We calculated the LasR-binding DNA motif using sequences near SNPs in the promoter-like and nonpromoter-like regions. A position weight matrix (PWM) model generated based on the promoter sequences was close to previously established results using chromatin immunoprecipitation (Fig. 3K) (15). This reverse complementary PWM model reflected the reverse-symmetric dimeric binding of LasR to its recognition box. In contrast, the PWM model generated based on those nonpromoter sequences partially reproduced features of the former motif, while losing the base-enrichment information for several locations in the PWM model (Fig. 3K). From an evolutionary perspective, TF-binding boxes in the intergenic regions at the 5' end of genes are likely to be subject to natural selection, and are closer to the best-fit box, thus facilitating TF binding and transcriptional regulation. While TF-binding boxes found in coding sequences are likely to appear in random, owing to the large proportion of coding sequences in the bacterial genome, most of these boxes only partially fit the TF-binding motif. Therefore, we calculated TF-binding motifs using sequences extracted from promoter-like regions in our subsequent experiments.

We validated the newly found LasR-binding promoters using electrophoretic mobility shift assay (EMSA). 10 of the 12 tested promoters exhibited noncooperative binding with LasR (15), while the remaining two promoters (*PA3347*, *PA5454*) exhibited cooperative binding patterns (Fig. 4A). These results demonstrate the success of AIDmut-Seq in detecting LasR-binding sites across the genome. Interestingly, real-time RT-PCR results confirmed LasR as a repressor for most of the newly found promoters, with a 2- to 3-fold increase in transcription levels after *lasR* deletion (Fig. 4B). LasR was generally considered to be a transcriptional activator in *P. aeruginosa*, while these data suggest that it directly represses the transcription of many genes.

Furthermore, we checked the necessity of knocking out the native TF genes in the bacterial genome. The native *lasR* gene was retained and AID732-LasR was allowed to initiate mutations in the *lasI*, *rhlI*, and *ung* triple mutant strain. A similar mutation spectrum was obtained as the result from the *lasI*, *rhlI*, *ung*, and *lasR* quadruple mutant strain, and the vast majority of SNPs were reproduced in the triple mutant strain (Supplemental File 1: Fig. S2). Therefore, the deletion of native TF genes had little effect on the output of AIDmut-Seq for LasR. However, due to the different intracellular levels of TF proteins, the above results for LasR may not be the case for other TFs. Nevertheless, for simplicity, subsequent AIDmut-Seq experiments with other TFs were conducted without TF gene deletion, i.e., in the *ung* single-knockout PAO1 strain.

**AIDmut-Seq exhibits high efficiency for FleQ and ErdR.** FleQ is an c-di-GMP effector that regulates biogenesis of flagella and biofilm formation in *P. aeruginosa* (8, 17, 18). With the incorporation of a partner protein FleN, FleQ switches from a repressor to an activator for *cdrA*, *pelA*, and *pslA* transcription in response to c-di-GMP binding (8, 18). We identified 32 FleQ target promoters, in which 21 of them were reported previously, and 11 promoters were newly identified in this study (Supplemental File 4). In addition, 15 previously FleQ-targeted promoters identified from ChIP-Seq were devoid in our AIDmut-Seq results (7). We tested those newly found targets using EMSA and

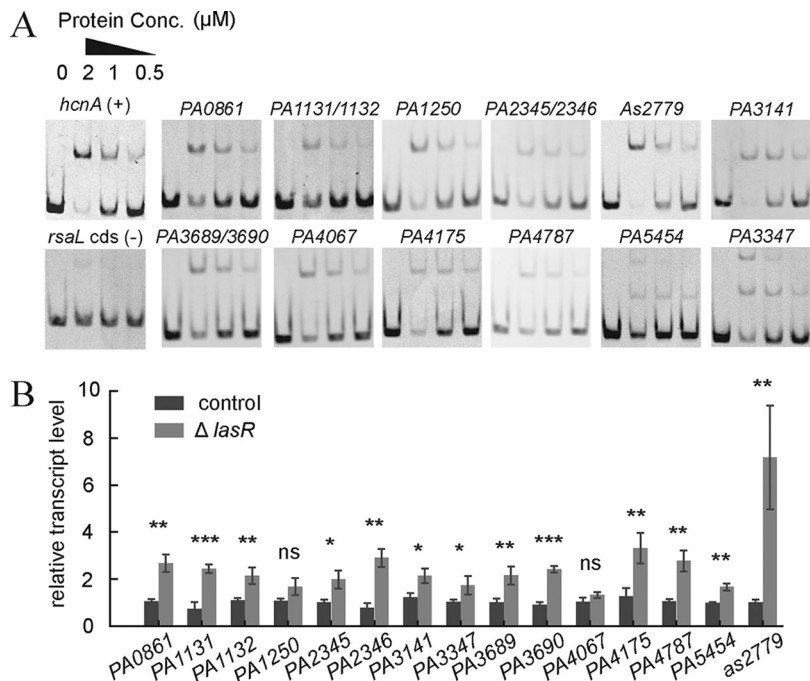

**FIG 4** Validation of the newly identified LasR-targeted promoters from the AIDmut-Seq experiment. (A) Validation of LasR binding to 12 of the newly identified promoters using EMSA. (B) Quantification of the transcription of LasR-targeted promoters in the control and the *lasR* mutant control strains. Control strain: Δ*lasI*Δ*rhlI* double mutant; and *lasR* mutant control strain: Δ*lasI*Δ*rhlI*Δ*lasR* triple mutant. Statistical analysis was carried out by two-sample *t* test. ns, nonsignificant; *, $P < 0.05$; **, $P < 0.01$; ***, $P < 0.001$.

real-time RT-PCR. Five promoters (*pvdP*, *infA*, *tlpQ*, *PA2955*, *lysP*) exhibited direct binding to FleQ, while the rest promoters have no binding affinity with FleQ (Fig. 5A). Transcripts of *flgF*, *PA1441*, *pvdP*, *PA2393*, *tlpQ*, and *PA2955* in the *fleQ* mutant are lower than those of the wild type (Fig. 5B), indicating that FleQ activates the transcription of these genes. Specially, the expression of the high-affinity histamine chemotaxis receptor gene *tlpQ* showed more than 40-fold decrease when *fleQ* was knocked out, which was not reported in previous studies. Transcripts of *PA0359* and PA4583 are slightly increased in the *fleQ* mutant, suggesting a weak transcriptional repression effects of FleQ on these two genes. Notably, neither did *fdx1* and *fimW* promoters bind to FleQ, nor did their expressions affected by *fleQ* deletion, indicating they are probably false-positive targets from AIDmut-Seq.

ErdR is a two-component response regulator that controls the expression of genes for ethanol oxidation, including the putative sensor kinase gene *ercS* and the response regulator gene *agmR* (19, 20). Previous study also demonstrated that ErdR mediates the expression of the *P. aeruginosa* acetyl-CoA synthetase gene *acsA* when acetate or ethanol was served as the carbon source (20). Currently, the direct regulon of ErdR is undetermined. We measured the genomic targets of ErdR using AIDmut-Seq, in which bacteria were cultured in a minimal FAB medium (21) supplemented with acetate as the sole carbon source. Fifteen promoters were identified, including the *acsA* promoter, but not *ercS* and *agmR* promoters (Fig. 5C and D, Supplemental File 4). EMSAs validated the binding of 13 of these promoters to phosphorylated ErdR (treated by acetyl phosphate) (Fig. 5D). We quantified the transcription level of these genes in the wild type and the *erdR* mutant strain under an acetate-induced growth condition. Except *PA3235*, most of the tested genes exhibited increased expression when *erdR* was deleted (Fig. 5C), indicating that ErdR represses the transcription of these promoters. Notably, expression of *PA3235*, which forms an operon with *PA3234*, decreased about 2-fold in *erdR* mutant. PA3234 (YjcG) has 82% sequence identity with the cation/acetate symporter ActP in *Escherichia coli*. The

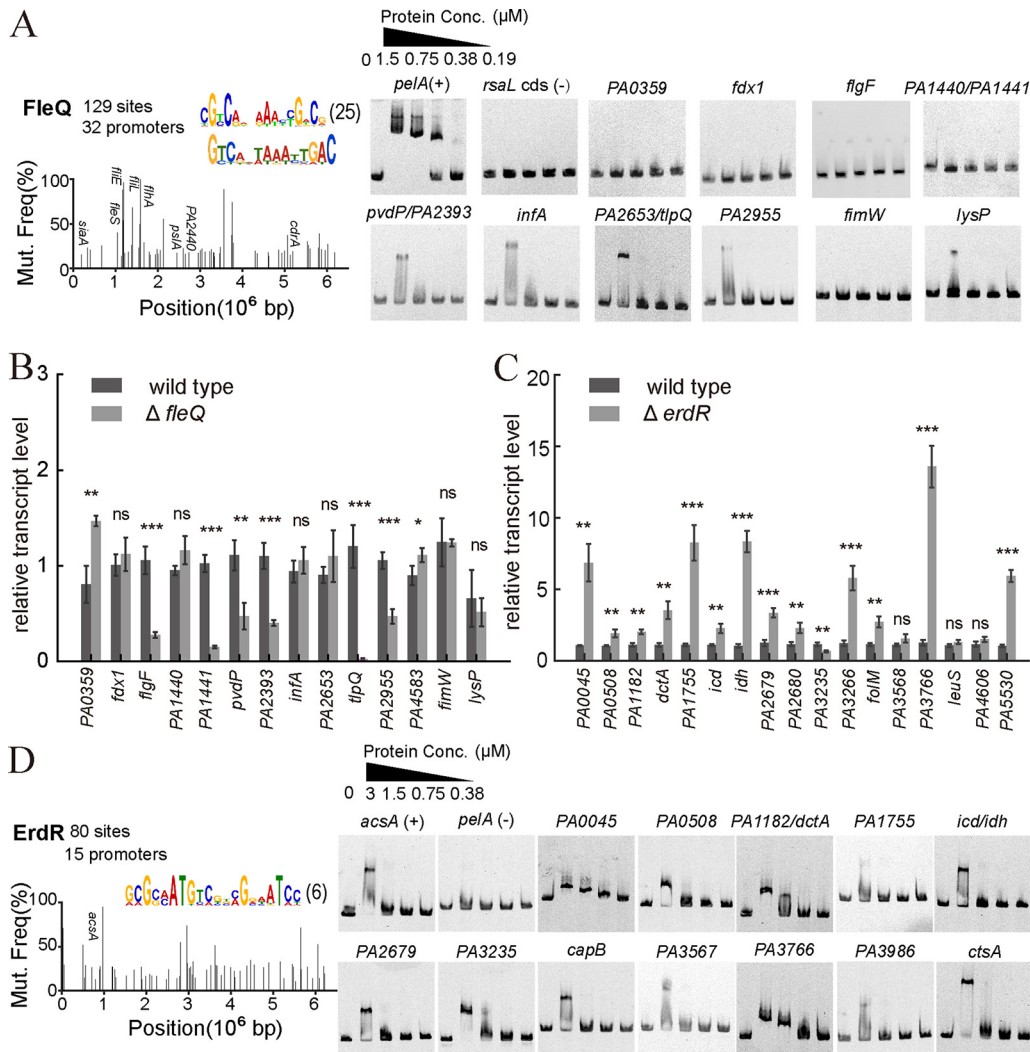

**FIG 5** Application of the AIDmut-Seq method to FleQ and ErdR in *P. aeruginosa*. (A) SNP spectrum obtained from the AIDmut-Seq result for FleQ (left), and the validation of FleQ binding to newly identified promoters using EMSA (right). Calculated TF-binding DNA motifs based on the AIDmut-Seq data are presented at the top right of each SNP spectrum. The upper motif was generated according to AIDmut-Seq data, lower motif represents the previously established PWM model. The number in parentheses after motifs indicates the number of promoters used to calculate the motif. Promoters known to be regulated by FleQ were annotated in the mutation spectra. (B) Quantification of the transcription of FleQ-targeted promoters in the wild type PAO1 and the *fleQ* mutant strains. (C) Quantification of the transcription of ErdR-targeted promoters in the wild-type PAO1 and the *erdR* mutant strains. Statistical analysis was carried out by two-sample *t* test. ns, nonsignificant; *, $P < 0.05$; **, $P < 0.01$; ***, $P < 0.001$. (D) SNP spectrum obtained from the AIDmut-Seq result for ErdR (left), and the validation of ErdR binding to newly identified promoters using EMSA (right).

above result suggests that ErdR triggers bacterial acetate intake through the regulation of YjcG expression. Other genes whose expression were significantly affected by *erdR* deletion includes the putative acyl-CoA dehydrogenase gene *PA0508*, the C4-dicarboxylate transporter gene *dctA*, isocitrate dehydrogenase genes *icd* and *idh*, the putative quinone oxidoreductase gene *PA2680*, the folate biosynthesis gene *folM*, the putative aromatic amino acid transporter gene *PA3766*, and the C5-dicarboxylate transporter gene *PA5530*, indicating the multiple routes of ErdR in regulating carbon metabolism.

We also tested the efficiency of AIDmut-Seq using the major type III secretion system activator ExsA and the two-component regulator GacA in *P. aeruginosa* (22–24). Half of the previously confirmed downstream promoters of ExsA were identified, including the *exsC*, *exsD*, *exoS*, and *popN* promoters (Supplemental File 1: Fig. S3, Supplemental File 4). While some expected ExsA targets were not found, such as the *exoT*, *exoY*, *impA* promoters (22, 25).

We also found five new potential ExsA target promoters. GacA is a primary regulator that participates in *P. aeruginosa* posttranscriptional regulation, through its activation of the expression of two small RNAs, RsmY and RsmZ (23). We reconfirmed the two known GacA-dependent promoters *rsmY* and *rsmZ* in the mutation spectrum, and two potential new GacA targets were also found (Supplemental File 1: Fig. S3, Supplemental File 4). Currently, the exact reason that causes the false-negative and false-positive results of FleQ and ExsA is undetermined, possible involved factors are listed in the discussion section. Nevertheless, the above results demonstrate the value of AIDmut-Seq for various transcriptional activators.

**AIDmut-Seq identifies only small fraction of known genomic targets for RsaL and AmrZ.** We then tested three transcriptional repressors GntR, HmgR, and RsaL, and one dual transcriptional regulator AmrZ. GntR directly represses the transcription of *gntR* and *gntP* promoters in *P. aeruginosa*, and the repression is released by binding of gluconate or 6-phosphogluconate to promoter bound GntR (26). HmgR binds and represses transcription of *hmgA* promoter unless it is bound to its ligand homogentisate in *Pseudomonas putida* (27). The HmgR homolog gene and its putative binding site in the *hmgA* promoter were also mapped in *P. aeruginosa* (27). AIDmut-Seq successfully identified the reported promoter targets for GntR (*gntR*, *gntP*) and HmgR (*hmgA*) (Fig. 6A). Notably, these are all the currently known targets for GntR and HmgR, and no extra promoter targets were found for the two regulators by AIDmut-Seq, suggesting that GntR and HmgR have few genomic targets in *P. aeruginosa*.

RsaL was initially identified as a repressor for *lasI* transcription and provides quorum sensing homeostasis in *P. aeruginosa* (28, 29). Later studies confirmed that it acts as a global regulator on bacterial virulence and directly binds promoters of genes required for the *Pseudomonas* quinolone signal (PQS) and pyocyanin biosynthesis (30, 31). Using AIDmut-Seq for RsaL, we identified the *lasI* promoter, but not the previously verified *cpdR*, *PA2228*, *phzM*, *hcnA*, and *phzA1* promoters. Besides, eight new potential promoter targets were identified, including promoters of the bacterial swarming regulator gene *bswR*, and the small RNA gene *reaL* modulating PQS synthesis (32, 33). Interestingly, although large number of known targets for RsaL were devoid in our AIDmut-Seq results, we successfully mapped RsaL binding sites on six of the newly found promoters (Supplemental File 1: Fig. S4A). And the identified promoters were able to generate a RsaL-binding DNA motif that quite similar to the former determined one (Fig. 6A) (30). These results increase the confidence of AIDmut-Seq data.

The alginate and motility regulator AmrZ is a dual-functional regulator that represses the transcription of *fleQ*, *amrZ*, and *psl* operon, while activates *algD* transcription (34–36). A previous study using ChIP-Seq combining with RNA-Seq revealed an AmrZ direct regulon containing hundreds of promoters, including 89 AmrZ-activated promoters (*pelB*, *algD*, *cdrA*, etc.) and 249 AmrZ-repressed promoters (*adcA*, *amrZ*, *fleQ*, *flgG*, *fliF*, etc.) (37). We applied AIDmut-Seq to AmrZ. Unfortunately, only six known targets were detected (*PA1913*, *pslA*, *amrZ*, *PA3722*, *PA4684*), other AmrZ-binding sites previously observed from ChIP-Seq were missing. Similar to the case of RsaL, we obtained an AmrZ-binding DNA motif very close to the previously reported PMW model, and AmrZ binding sites were mapped in three of the newly identified promoters (Supplemental File 1: Fig. S4B). Therefore, AIDmut-Seq identified some reliable but only a small fraction of genomic targets for both RsaL and AmrZ.

RsaL and AmrZ are both small TFs with molecular weights of 12.3 kDa and 9.4 kDa. The fusion of the 24 kDa AID732 to these small TFs may interfere with their proper folding, thus reducing their ability to bind to target genomic sites. In addition, RsaL and AmrZ mainly act as transcriptional repressors in *P. aeruginosa*. Overexpression of them can lead to very low transcription rate of some genes, thereby precludes AID732 accession to ssDNA which is dependent on the formation of transcription open complex. Therefore, when using AIDmut-Seq for transcriptional repressors, the intracellular level of AID732-TF fusion proteins needs to be further optimized, so that the chimeric TFs are sufficient to bind their genomic targets while not overexpressed to disable the transcription of target promoters.

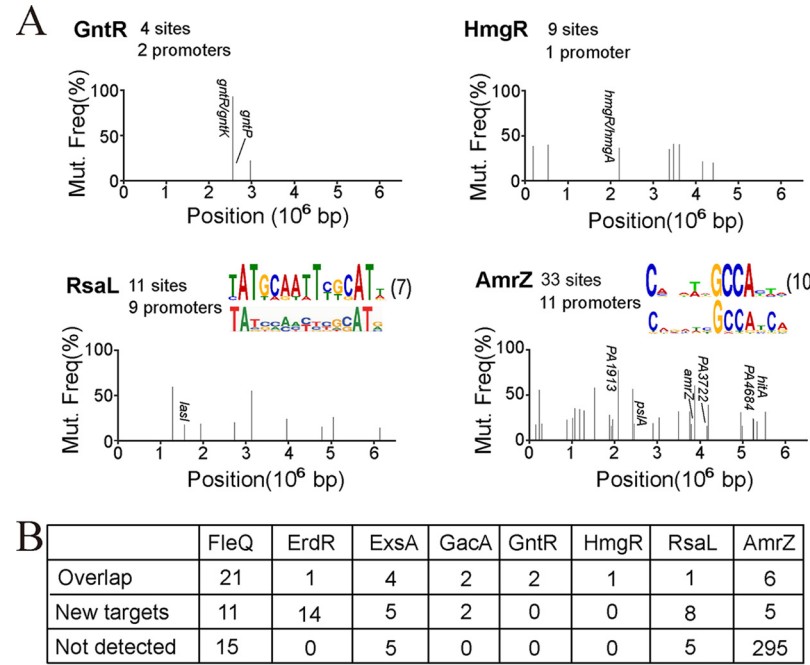

**FIG 6** Application of the AIDmut-Seq method to transcriptional repressors in *P. aeruginosa*. (A) SNP spectrum of GntR, HmgR, RsaL, and AmrZ. Calculated TF-binding DNA motifs based on the AIDmut-Seq data are presented at the top right of RsaL and AmrZ panels, in which the upper motifs were generated according to AIDmut-Seq data, lower motifs represent the previously established PWM models. The numbers in parentheses after motifs indicate the number of promoters used to calculate the motif. Promoters known to be regulated by those TFs were annotated in the mutation spectra. DNA motifs of GntR and HmgR were not obtained because of few identified promoters. (B) Comparison of AIDmut-Seq results with previously reported TF-targets. The statistic considers only the promoter-like intergenic targets identified by AIDmut-Seq. Overlap, number of AIDmut-Seq identified TF-targets that were previously reported. New targets, number of AIDmut-Seq identified TF-targets that were not reported previously. Not detected, number of previously reported TF-targets that were not identified by AIDmut-Seq.

**Benchmarking of AIDmut-Seq with ChIP-seq.** ChIP-seq is the most commonly used method for genome-wide mapping of TF targets in bacteria. We then compared previously published ChIP-seq data (7, 37, 38) of several TFs (GacA, ExsA, FleQ, AmrZ) with the corresponding AIDmut-Seq results. AIDmut-Seq and ChIP-seq detected similar numbers of target promoters in PAO1 genome for GacA, ExsA, and FleQ, while ChIP-seq detected far more targets for AmrZ than AIDmut-Seq (Fig. 7A).

Mutation sites detected by AIDmut-Seq located within 10 to 100 bp from TF binding boxes, while ChIP-seq peak summits located within a wider distance window, 1 to 1,000 bp from TF-binding boxes (Fig. 7B). The mean and median distance of AIDmut-Seq mutation sites to TF-binding boxes is 35 and 25 bp, respectively, and the corresponding value for ChIP-seq peak summits are 62 and 14 bp, respectively (Fig. 7C). Thus, AIDmut-Seq and ChIP-seq have similar detecting accuracies for TF-binding sites.

AIDmut-Seq mutations exhibited two patterns for different TFs in the promoter region. For GacA, ExsA, and AmrZ downstream promoters, only one mutation site or one mutation active region with successive mutations spaced several bases apart was detected (Fig. 7D, Supplemental File 3). Whereas for many downstream promoters of FleQ, two or more mutation sites were detected. This is probably arisen from the multiple FleQ binding boxes within these promoters, as exemplified by promoters of *pslA*, *PA2440*, *PA3177*, *PA3340*, and *cdrA* (Fig. 7D). However, ChIP-seq data exhibited only one peak for all promoters. Thus, AIDmut-Seq has a potential advantage to identify multiple TF-binding boxes in a promoter.

## DISCUSSION

In this study, we have developed an easy, *in vivo*, effective method for mapping the genomic targets of TFs and validated its feasibility in *P. aeruginosa*. The AIDmut-Seq

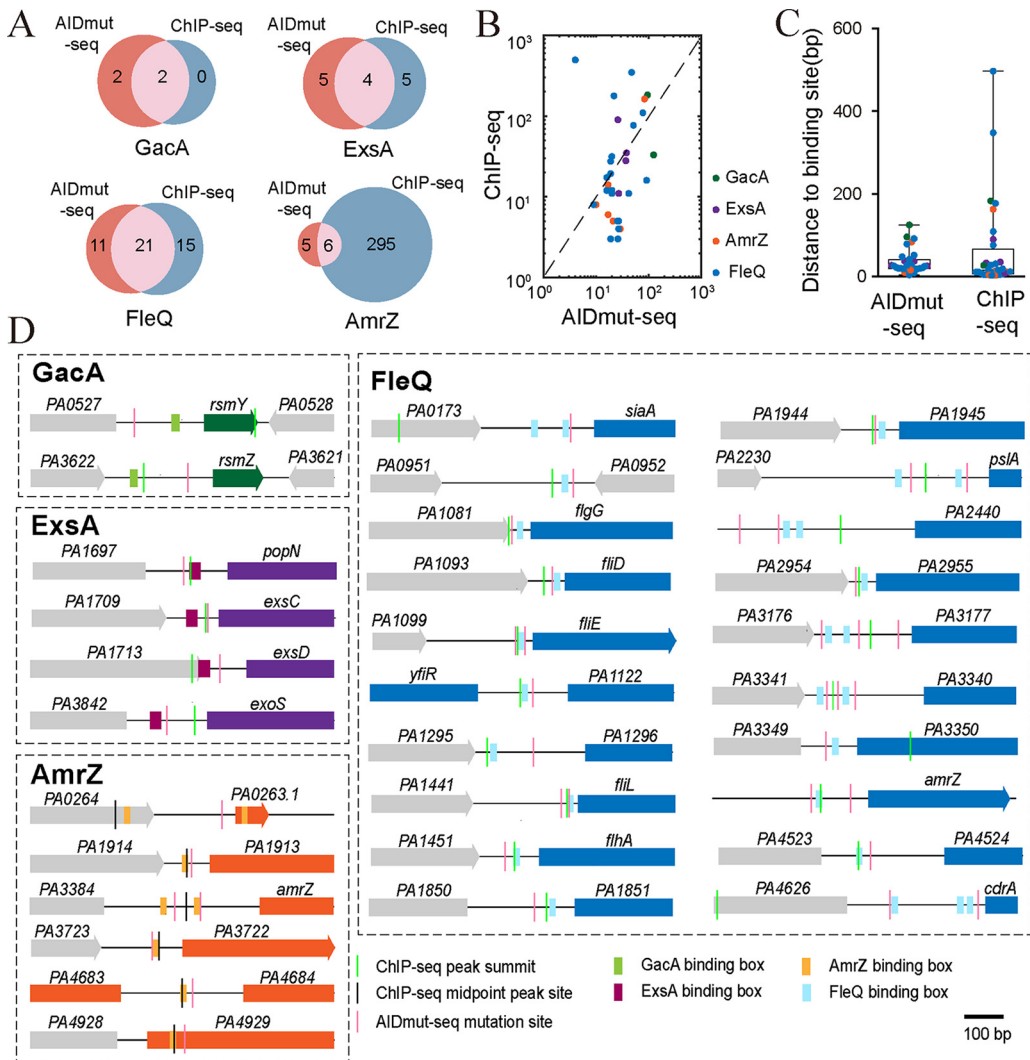

**FIG 7** Benchmarking AIDmut-Seq with ChIP-seq. (A) Venn diagram for GacA, ExsA, AmrZ, and FleQ responsive targets detected by AIDmut-Seq and targets detected by ChIP-seq. (B) Bilogarithmic plot of distance from ChIP-seq peak centers to TF-binding boxes (Y-axis) against distance from AIDmut-Seq mutation sites to TF-binding boxes (X-axis). The midpoint position of ChIP-seq peaks were used instead of positions of ChIP-seq peak centers for AmrZ. (C) Average distance between TF-binding boxes to AIDmut-Seq mutation sites or ChIP-seq peak centers. Boxes indicate median and 25% to 75% of all data points, whiskers indicate range. (D) Schematic depicting of the genome locations of AIDmut-Seq mutations, ChIP-seq peak centers, and TF-binding boxes for GacA, ExsA, AmrZ and FleQ. AIDmut-Seq mutation sites were represented as pink lines, ChIP-seq peak centers were represented as light green lines, ChIP-seq midpoint peak sites were represented as dark lines.

method consists of three steps: (i) the expression of the AID732-TF fusion protein in the *ung* mutant; (ii) the extraction of genomic DNA and whole-genome sequencing; and (iii) SNP profiling. The whole procedure requires only basic techniques in molecular biology and can be finished within 1 month, thus facilitating its use by junior investigators or interdisciplinary researchers who are not familiar with complicated biological experiments.

The false-negative results from AIDmut-Seq may because of the following reasons: (i), the fusion of AID732 to TFs precludes the binding of TFs to some target sites, which is probably the case in AmrZ and RsaL; (ii) AID732-initiated base conversions require its access to ssDNA, which is mainly provided in the forms of transcription initiation bubble, and some targets may have very low transcription initiation rates; (iii) the neighboring sequences of some targets are less susceptible to mutation by AID732; (iv) the expression of native TFs have a considerable competitive binding with some high-affinity targets; (v) some targets may have too low affinity to their corresponding TFs, resulting in a

short dwelling time of TF-binding state, which is not enough for AID732 to initiate deamination. The exact reasons that cause the occurrence of false-negative results may differ case by case, which is not determined currently. To address this issue, further studies regarding the effect of protein fusion strategy, transcription activity measurement of target promoters, sequence bias of AID732, native TF expression, and TF-binding dynamics are needed.

The false-positive results from AIDmut-Seq, as exemplified by EMSA results of FleQ, may arise from (i) the nonspecific binding of TFs, which occurs when intracellular TF level is high; (ii) stochastic genome mutations during strain construction; and (iii) TF-guided AID732 deamination of bases that are adjacent to TF targets in 3D-space. To eliminate those false-positive results, further efforts regarding the control of protein expression level, the optimization of experimental procedure to avoid stochastic mutations, and the replacement of different fusion linkers are required.

The defects of AIDmut-Seq are obvious, elimination of those false-negative and false-positive results are needed in further studies. However, preliminary results have demonstrated the efficiency of AIDmut-Seq, which can serve as a complementary tool for existing methods.

## MATERIALS AND METHODS

**Strains and media.** All AIDmut-Seq experiments used the *P. aeruginosa* strain PAO1: Δ*ung*, Δ*lasI*Δ*rhlI*Δ*lasR*Δ*ung*, and Δ*lasI*Δ*rhlI*Δ*ung*, Δ*fleQ*, Δ*erdR*, *etc*. All gene knockout mutants were obtained using a markless gene deletion protocol. (39) Detailed strain information is provided in the Supplemental File 1 (Table S1). Strains carrying AID-TF fusion plasmids were grown in the classical LB broth (10 g/L NaCl, 10 g/L tryptone, 5 g/L yeast extract) supplemented with 30 $\mu$g/mL gentamicin (LB+gen30), and 20 $\mu$M 3OC12-HSL (sigma Aldrich, cat. number O9139), or 0.4% (w.t.) arabinose (Sangon Biotech, cat. number A610071) was used when induction of AID-TF expression was required. For the AIDmut-Seq experiment of ErdR, bacteria were cultured in the FAB medium supplemented with 30 mM acetate and 30 $\mu$g/mL gentamicin, 0.4% (w.t.) arabinose was used when induction of AID-TF expression was required.

**Construction of plasmids.** All AID-TF fusion constructs were derived from pJN105, harboring an arabinose-inducible expression system. TF gene fragments without the stop codon were amplified from the PAO1 genome using PCR, and codon-optimized AID gene variants were synthesized by Sangon Biotech (Shanghai, China). The TF gene, AID, and the linkers between them were cloned into pJNl05 via Gibson Assembly (Vazyme, ClonExpress MultiS one Step Cloning kit. cat. number C113-02), immediately after the pBAD promoter. The exact sequences of AID variants, linkers (Table S2), and pJN105 vector are all included in Supplemental File 1. Plasmids were electroporated into the PAO1 strains. Colonies were selected on LB agar (1.5% w.t.) plates supplemented with 30 $\mu$g/mL gentamicin and verified via PCR and sequencing. Plasmids and PCR primers used in this study are included in Table S1 of Supplemental File 1.

**Estimation of mutation frequency via first-generation sequencing.** DNA fragments (about 500 bp) of the LasR-responsive promoters (*rhlI*, *lasB*, *rsaL*, *PA3904*, *ambB*, and *PA4677*) were amplified from the genome of AID-LasR fusion strains with different fusion directions, AID variants, linker types, and culture conditions. The PCR products were sequenced using first-generation sequencing, and the mutation frequencies were calculated using the Mutation surveyor software (40).

**AIDmut-Seq sample preparation and sequencing.** Bacterial strains carrying AID-TF expression vectors were recovered from −80℃ fridge on LB agar (1.5% w.t.) plates supplemented with 30 $\mu$g/mL gentamicin and incubated for 16 h at 37℃. Bacteria were scraped from plates and resuspended in 1 mL of fresh LB+gen30 media to $OD_{600}$∼1.0. Then, inducers were added (3OC12-HSL+arabinose for LasR-related experiments, arabinose alone for other experiments), and the cultures were shaken for 12 h at 37℃. The cultures were further diluted five times into LB+gen30 with inducer and cultured for another 12 h. Bacteria were harvested via centrifugation, and the genomic DNA was extracted using a commercial kit (Tiangen Biotech, Beijing, China; cat No. DP302) following the manufacturer's instructions. Whole-genome sequencing was conducted using Illumina NovaSeq 6000 instruments in Sangon Biotech. See Supplemental File 1 for details.

**SNP profiling and generation of TF-binding DNA motifs.** SNPs measured in the genome of PAO1 wild-type strain (caused by strain transfers from lab to lab, Supplemental File 5) were excluded first from all the data obtained from AIDmut-Seq experiments. Then, common SNPs measured in three parallel experiments of the pJN (PAO1Δ*ung* carrying pJN105 void vector) and AID732pJN (PAO1Δ*ung* carrying AID732-pJN105) strains were identified, and combined to generate a union SNP group, represented as SNP$^{negative}$. For each TF, the common SNPs were obtained from three parallel AIDmut-Seq experiments in the AID732TF strain, represented as SNP$^{ori}$. TF-guided SNPs were then obtained by excluding SNPs common to SNP$^{negative}$ from SNP$^{ori}$, and this set was represented as SNP$^{TF}$. We further deleted the SNPs of SNP$^{TF}$ in two cases: (i) SNPs were found in the pf4 prophage region from *PA0715* to *PA0729,* and (ii) SNPs that appeared more than twice in AIDmut-Seq experiments for different TFs. We consider that the probability of a promoter being regulated by three different TFs is very low as these SNPs were generated through weak-specific binding of TFs to DNA. The resulting SNP group was further divided into SNP$^{TF, promoter}$ and SNP$^{TF, nonpromoter}$, according to whether they were located in the promoter region of a gene. In the case of the SNP profiling for LasR in

the $\Delta lasI \Delta rhlI \Delta ung \Delta lasR$ strain, SNPs in the above-mentioned pJN and AID732pJN strain were replaced by SNPs in pJN_b1 and AID732pJN_b1.

To calculate the TF-binding DNA motifs using SNP$^{\text{TF, promoter}}$ or SNP$^{\text{TF, nonpromoter}}$, 100 bp of sequence flanking the upstream and downstream of each SNP site (201 bp in all) was retrieved and summarized into a text file. The DNA motifs for TF binding were obtained using the online MEME tool (41), with the motif wide set to 8 to 20 bp. SNP attributions and annotations, sequence retrieval, and profiling were performed using code developed in-house in MATLAB. The DNA sequence and annotations of the PAO1 genome were downloaded from the *Pseudomonas* database (www.pseudomonas.com). The reference TF-binding motifs for LexA and ExsA were redrawn from data in the CollecTF database (http://www.collectf.org).

**Protein purification and electrophoretic mobility shift assay.** LasR, ErdR, and FleQ purification was performed following protocols based on Nickel NTA column. For details, see the Supplementary Methods in Supplemental File 1. EMSAs were performed using 200 bp DNA fragments. TF protein of different concentrations were mixed with 30 ng DNA and incubated at room temperature for 30 min in binding buffer (20 mM Tris-HCl pH = 7.8, 50 mM KCl, 1 mM EDTA, 10% glycerol), and 100 $\mu$M 3OC12-HSL was added to all DNA-LasR binding reactions, 0.2 mM acetyl-phosphate was added to all DNA-ErdR binding reactions. Gel electrophoresis was run in $0.5\times$ TBE buffer at 100 V for 90 min. The DNA bands in the gels were stained with $10000\times$ GelRed for 40 min and imaged using a UVP Chemstudio touch instrument (Analytik Jena, Jena, Germany). PCR primers used to amplify promoter segments are included in Table S1 of Supplemental File 1.

**Real-time RT-PCR experiment.** The $\Delta lasI \Delta rhlI$ double mutant and the $\Delta lasI \Delta rhlI \Delta lasR$ triple mutant stains were used for pairwise comparison in LasR experiments. Wild type PAO1 and the *fleQ* mutant strains were used for pairwise comparison in FleQ experiments. Wild type PAO1 and the *erdR* mutant strains were used for pairwise comparison in ErdR experiments. Bacterial culture conditions were the same as that in AIDmut-Seq experiments for LasR and FleQ. For ErdR experiments, bacteria scraped from LB agar plate were resuspended into FAB medium supplemented with 30 mM glutamate + 30 mM acetate to OD~1.0 and were cultured (220 rpm shaking) for 12 h at 37°C before harvesting. Bacterial cells were harvested using centrifugation at 10000 rpm for 2 min. RNA purifications were conducted with a RNeasy minikit (Qiagen) and cDNA were generated from the extracted mRNA with a FastKing RT kit (Tiangen Biotech). RT-qPCR was performed by Hieff UNICON Universal Blue qPCR SYBR green Master Mix (YEASEN) according to the manufacturer's instruction. Each reaction was performed in triplicate in 20 $\mu$L reaction volumes with 600 ng cDNA and 50S rRNA as an internal control. In each reaction, 200 nM RT-qPCR primers were used. RT-qPCRs were run at 95°C for 2 min (predenaturation, 1 cycle), 95°C for 10 s and 60°C for 30 s (40 cycles). Real-time PCR primers are included in Table S1 of Supplemental File 1.

**Data availability.** The data for mutation spectra drawing are included in the Supplemental File 3. Computer codes for retrieval of sequence flanking the mutation sites are uploaded at GitHub website (https://github.com/Carl-Ni/Computer-code-for-sequence-retrieval.git).

## SUPPLEMENTAL MATERIAL

Supplemental material is available online only.

**SUPPLEMENTAL FILE 1**, PDF file, 0.7 MB.
**SUPPLEMENTAL FILE 2**, XLSX file, 0.02 MB.
**SUPPLEMENTAL FILE 3**, XLSX file, 0.03 MB.
**SUPPLEMENTAL FILE 4**, XLSX file, 0.01 MB.
**SUPPLEMENTAL FILE 5**, XLSX file, 0.01 MB.

## ACKNOWLEDGMENTS

We thank Jia Ning's lab at the Southern University of Science and Technology for giving important advice and providing experimental equipment for the extraction of TF proteins. This work was supported by National Key Research and Development Program of China (Grant No. 2020YFA0906900 and Grant No. 2018YFA0902700 to Fan Jin) and the Scientific Instrument Developing Project of the Chinese Academy of Sciences (Grant No. YJKYYQ20200033 to Fan Jin).

Conceptualization, Lei Ni and Fan Jin; experiments, Feixuan Li, Xiao-Yu Liu; data analysis and investigation, Feixuan Li, Lei Ni; writing—original draft—Lei Ni; writing—review and editing—Fan Jin.

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
