## [Reviewer comments · Microbiology Spectrum]

Microbiology Spectrum

AIDmut-seq: A three-step method for detecting protein-DNA binding specificity

Fan Jin, Feixuan Li, Xiao-Yu Liu, and Lei Ni

Corresponding Author(s): Fan Jin, Chinese Academy of Sciences Shenzhen Institutes of Advanced Technology

Review Timeline:

Submission Date:	September 16, 2022
Editorial Decision:	November 13, 2022
Revision Received:	November 30, 2022
Accepted:	December 1, 2022

Editor: Beile Gao

Reviewer(s): The reviewers have opted to remain anonymous.

Transaction Report:

DOI: <https://doi.org/10.1128/spectrum.03783-22>

November 13, 2022

Dr. Fan Jin
Shenzhen Institute of Synthetic, Shenzhen Institutes of Advanced Technology Chinese Academy of Sciences
Shenzhen
China

Re: Spectrum03783-22 (AIDmut-seq: A three-step method for detecting protein-DNA binding specificity)

Dear Dr. Fan Jin:

Link Not Available

Sincerely,

Beile Gao

Journals Department
Reviewer comments:

Reviewer #1 (Comments for the Author):

This study describes the development and validation of a new method for the identification of transcriptional factors (TF) binding sites in bacterial genomes. *Pseudomonas aeruginosa* was used as model organism.

The method, called AIDmut-seq, is based on the arabinose-dependent expression of the TF fused to the activation-induced cytidine deaminase (AID) enzyme. The two proteins are divided by a flexible linker. During bacterial growth in the presence of arabinose, AID should make C-T or G-A conversions in the DNA sequence near each TF binding site in the target bacterial genome. The target genome should be deleted in the specific TF gene and, most importantly, also in the *ung* gene (coding for uracil-N-glycosylase), in order to impair the repair of AID-induced mutations. The latter are ultimately detectable by whole genome sequencing.

In the first part of the work, the LuxR-like activator LasR was used as model TF to set-up the method and define fusion protein

expression level, AID variant, linker variant, fusion direction etc.

In the second part of the work, the method was used to identify binding sites of other *P. aeruginosa* TFs. In particular, three activators (ExsA, GacA, ErdR) and three repressors (RsaL, GntR, HmgR) were taken into consideration.

The overall text is too concise, to the detriment of clarity. In fact, several passages are not clear (only some examples are given in the specific comments below). Spectrum research articles have not words or references limit; hence things can be better explained. The underlying rationale of the work is good, and the results obtained with LasR as model TR are convincing enough. The second part of the work is weak (see specific comments below).

Finally, there are a lot of minor comments, only some of these are reported below.

1) lines 1-63 - the basic principles underlying each cited method (e.g. Chip-seq; DAP-seq; SELEX; DamID; Calling cards) should be concisely but clearly explained. The differences between in vitro and in vivo methods should be highlighted. Authors state that the proposed method is feasible also for "junior investigators" and "interdisciplinary researchers"; hence, also a researcher not very expert in molecular biology should be able to understand and appreciate advantages and disadvantages of each method in comparison with the proposed one.

2) Experiment of figure 2, Authors should indicate which arabinose concentration was used and AID Δ expression level should be determined. Lines 118-120 and Lines 163-170, these paragraphs are not clear enough, please reformulate/explain better.

3) Experiment of figure 3; Figure 3L, newly identified LasR binding sites should be validated also in vivo by using transcriptional fusions and/or Real Time RT-PCR. This panel should be a separated figure.

4) Results obtained with the three repressor proteins (RsaL, GntR, HmgR) clearly indicate that this method is not appropriate for the detection of DNA binding sites of repressors (e.g. see the summary in figure 4C). Authors do not highlight properly this important result.

As also stated by Authors (lines 114-115 and 274-276), the AID enzyme works on single strand DNA that is originated during the formation of the transcription open complex. Hence, it makes sense that the method cannot work well for transcriptional repressors. This should be clearly discussed.

5) Among these repressors, RsaL is perhaps the most well-known and several papers with EMSA assays have been published. Authors should cite the papers produced between 2005 and 2009 by Giordano Rampioni et al.

6) Results obtained with the activator and dual TRs are not fully convincing. In particular, control EMSA assays have been provided only for the dual TR FleQ (Figure S3). EMSA should be provided also for ExsA, GacA, ErdR, AmrZ. In addition, transcriptional fusion experiments or RT-PCR experiments should be carried out to validate in vivo the EMSA results. Concerning FleQ, only half of the tested promoters showed a clear band-shift after binding of FleQ (PA2393, PA2619, PA2653, PA2955, PA4981). I understand that this is a lot of work, perhaps authors could limit their work to FleQ and another transcriptional activator.

Minor comments (partial revision, only up to line 137).

Line 64, please explain Ugi gene function and why this is important;

Lines 64-68- not very clear, please reformulate;

Line 71, AID from which organism?;

Line 73, what do you mean for toxic? This is not clear enough, please reformulate, explain better;

Lines 75-76, What is exactly the "dCas9-guided MS2-AID" mutation generator? It is likely that many readers could not know what the authors are talking about;

Line 76, please use instead of "will" use "could";

Line 85, please use "could cost" instead of "costs";

Line 88, what is exactly AID Δ ? What is the difference compared to the wild type AID?

Line 95, The LasR inducer is named N-3-oxo-dodecanoyl-homoserine lactone and should be abbreviated as 3OC12-HSL;

Line 105, please add reference for the tested promoters;

Line 137, not very clear, please clarify. Perhaps authors wants to say that "high expression levels of AID732-TF may lead to the detection of a large number of binding site with weak binding affinity".

Reviewer #2 (Comments for the Author):

In this manuscript, the authors developed a new method, called AIDmut-seq, to identify binding sites of transcriptional factor in vivo. The AIDmut-seq method is performed by three steps composing with fusion protein, extraction of genomic DNA and sequencing/SNP profiling. This approach is easy to be employed by junior and interdisciplinary investigators with only basic understanding in molecular biology. To establish the AIDmut-seq platform, the authors optimized the fusion direction, linker type,

AID variant, induction time and arabinose concentrations. After which, the AIDmut-seq was conducted to examine binding sites of quorum sensing regulator LasR. Sequencing depths and reproducibility of SNPs in independent experiments were compared to evaluate the repeatability AIDmut-seq. Further, AIDmut-seq was applied to other transcriptional factors in *P. aeruginosa* such as FleQ, AmrZ, ExsA and etc.. Subsequently, EMSA was conducted to validate detected target sequences. Finally, the authors compared AIDmut-seq method with the classic Chip-seq method, and showed that AIDmut-seq has several advantage compared to Chip-seq. In summary, this study developed a useful approach for detecting protein-DNA binding sites which might support research field about transcriptional regulatory network.

I think this manuscript should be properly revised before its acceptance.

Minor comments:

- 1) Line 20 and 35, "in vivo" should be written in italic.
- 2) Line 64, SNPs full name should be shown at first time.
- 3) Fig. 3A-3L should be reordered according to the order they appear in the text, as well as Fig. 4A-4C.
- 4) Manufacturer and affiliated states of reagents and kits in Methods should be provided.
- 5) Line 359, "*Pseudomonas*" should be written in italic.
- 6) A large number of species names are not italicized, such as *Pseudomonas aeruginosa* at line 491 in References. The format of some references is wrong. For example, the first letter of each word of paper title are capitalization. The authors need revise.
- 7) Strains, plasmids and primers used in this study are listed in supplemental material.
- 8) Line 25-26 and line 40 in supplemental material, "*P. aeruginosa*" and "*Escherichia Coli*" are revised in italic.
- 9) Line 40 in supplemental material, pET28a instead of pet28a.
- 10) Line 172 in supplemental material, LasR instead of lasR.

Staff Comments:

Preparing Revision Guidelines

Please return the manuscript within 60 days; if you cannot complete the modification within this time period, please contact me. If you do not wish to modify the manuscript and prefer to submit it to another journal, please notify me of your decision immediately so that the manuscript may be formally withdrawn from consideration by Microbiology Spectrum.

Re: Spectrum03783-22 (AIDmut-seq: A three-step method for detecting protein-DNA binding specificity)

The ASM Journals program strives for constant improvement in our submission and publication process. Please tell us how we can improve your experience by taking this quick Author Survey.

I

Reviewer comments:

Reviewer #1 (Comments for the Author):

This study describes the development and validation of a new method for the identification of transcriptional factors (TF) binding sites in bacterial genomes. *Pseudomonas aeruginosa* was used as model organism.

The method, called AIDmut-seq, is based on the arabinose-dependent expression of the TF fused to the activation-induced cytidine deaminase (AID) enzyme. The two proteins are divided by a flexible linker. During bacterial growth in the presence of arabinose, AID should make C-T or G-A conversions in the DNA sequence near each TF binding site in the target bacterial genome. The target genome should be deleted in the specific TF gene and, most importantly, also in the *ung* gene (coding for uracil-N-glycosylase), in order to impair the repair of AID-induced mutations. The latter are ultimately detectable by whole genome sequencing.

In the first part of the work, the LuxR-like activator LasR was used as model TF to set-up the method and define fusion protein expression level, AID variant, linker variant, fusion direction etc.

In the second part of the work, the method was used to identify binding sites of other *P. aeruginosa* TFs. In particular, three activators (ExsA, GacA, ErdR) and three repressors (RsaL, GntR, HmgR) were taken into consideration.

The overall text is too concise, to the detriment of clarity. In fact, several passages are not clear (only some examples are given in the specific comments below). Spectrum research articles have not words or references limit; hence things can be better explained. The underlying rationale of the work is good, and the results obtained with LasR as model TR are convincing enough. The second part of the work is weak (see specific comments below).

Finally, there are a lot of minor comments, only some of these are reported below.

Reply: We thank the reviewer for giving these important and helpful comments. We have revised the manuscript according to the reviewer's comments, including rewriting the abstract, introduction, and results section and supplementing several real time RT-PCR results in the main text. The second part of the work was reorganized. The major and minor comments were addressed point by point. We look forward to further suggestions from the reviewer.

1) lines 1-63 - the basic principles underlying each cited method (e.g. Chip-seq; DAP-seq; SELEX; DamID; Calling cards) should be concisely but clearly explained. The differences between in vitro and in vivo methods should be highlighted. Authors state that the proposed method is feasible also for "junior investigators" and "interdisciplinary researchers"; hence, also a researcher not very expert in molecular biology should be able to understand and appreciate advantages and disadvantages of each method in comparison with the proposed one.

Reply: Thanks for the comment. We have added detail descriptions of all the cited methods in the introduction section in our revised manuscript (page 3-7, line 57-67, line 79-96). In addition, we have added descriptions about the difference between in vivo and in vitro methods (page 4-5, line 71-78).

2) Experiment of figure 2, Authors should indicate which arabinose concentration was used and AID Δ expression level should be determined. Lines 118-120 and Lines 163-170, these paragraphs are not clear enough, please reformulate/explain better.

Reply: Thanks for the comment. The arabinose concentration we used is 0.4%, corresponding to an intracellular AID Δ expression of 7~8 μ M, as determined by quantification of SfGFP expression at the same arabinose concentration under microscope. We have added the information in the revised manuscript (page 8, line 148). In addition, we have reformulated the descriptions for line 118-120 and line 163-170 as below:

Line 118-120 was rewritten as (page 9, line 170-181): "Besides, mutations generated in the promoter or coding sequences of a gene can affect its expression level or result in a loss of gene function, both of which may reduce the growth rate of a cell, leading to a decreased proportion of the mutated cells in the whole population after multiple generations. This will reduce the detected mutation frequencies of some mutations. To minimize the impact of possible growth rate reduction on the detection of genomic mutations, we added the inducers at an initial OD_{600} of 1.0 and diluted bacterial culture 5 \times for each 12 hours of shaking. Using this approach, the bacteria can take several generations to complete the C-T conversions induced by AID Δ , while the population will not experience too many generations, which will eliminate those mutants with low growth rates. We observed the highest mutation frequency in all promoters with a culture time of 24 hours (two rounds of 12-hour culture)."

Line 163-170 was rewritten as (page 11-12, line 220-231): "To eliminate these false-positive SNPs, we extracted the shared mutations of the AID732LasR_b1 genome in three independent

experiments. According to our sequencing results of AID732pJN_b1, the average probability of stochastic mutation (p_0) generated by AID732 through the experimental procedure was less than 10^{-4} per base pair. We assume that AID732-LasR has a similar probability to generate stochastic mutation. Thus, the probability of one mutated base pair that occurs in both three independent experiments is p_0^3 . Then the average number of shared mutations from three independent experiments was $N \cdot p_0^3$. Here N is the total number of base pairs of the genome which is within the range of $10^6 \sim 10^7$ for common bacterial species. Therefore, $N \cdot p_0^3$ is far less than 1.0. That is, less than one shared stochastic mutation can be detected from three independent experiments. Thus, those stochastic false-positive results can be eliminated.”

3) Experiment of figure 3; Figure 3L, newly identified LasR binding sites should be validated also in vivo by using transcriptional fusions and/or Real Time RT-PCR. This panel should be a separated figure.

Reply: Thanks for the comment. We have moved Figure 3L, together with the supplemented real-time RT-PCR results of LasR-targeted promoters to a new Figure (Figure 4) in the revised manuscript.

Figure 4. Validation of the newly identified LasR-targeted promoters from the AIDmut-seq experiment. (A), Validation of LasR binding to twelve of the newly identified promoters using EMSA. (B) Quantification of the transcription of LasR-targeted promoters in the control and the *lasR* mutant control strains using real-time RT-PCR. Control strain: $\Delta lasI \Delta rhII$ double mutant; and *lasR* mutant control strain: $\Delta lasI \Delta rhII \Delta lasR$ triple mutant. Statistical analysis was carried out by two-sample t-test.

ns, non-significant; *, $p < 0.05$; **, $p < 0.01$; ***, $p < 0.001$.

4) Results obtained with the three repressor proteins (RsaL, GntR, HmgR) clearly indicate that this method is not appropriate for the detection of DNA binding sites of repressors (e.g. see the summary in figure 4C). Authors do not highlight properly this important result. As also stated by Authors (lines 114-115 and 274-276), the AID enzyme works on single strand DNA that is originated during the formation of the transcription open complex. Hence, it makes sense that the method cannot work well for transcriptional repressors. This should be clearly discussed.

Reply: Thanks for the comment. We agree with the reviewer that AIDmut-seq has some defects when applying to transcriptional repressors, as exemplified by the results of RsaL and AmrZ. For GntR and HmgR, we identified two and one of their known targets on the *P. aeruginosa* genome, which represent all of their known genomic targets. Hence, it is not certain whether there are additional targets of GntR and HmgR that were missed by AIDmut-Seq. In fact, we are currently considering that there should be a proper window of the intracellular expression level of AID732-TF fusion proteins. In this window, the chimeric TFs are sufficient to bind their genomic targets while not overexpressed to disable the transcription of target promoters. As suggested by the reviewer, we have collected AIDmut-seq results for transcriptional repressors in a separate part of the results section (and in Figure 6), and we discussed the possible reasons that cause failure of AIDmut-seq in identifying RsaL and AmrZ targets (page 19, line 382-391).

5) Among these repressors, RsaL is perhaps the most well-known and several papers with EMSA assays have been published. Authors should cite the papers produced between 2005 and 2009 by Giordano Rampioni et al.

Reply: Thanks for this kind suggestion. We have cited these papers in the revised manuscript (page 17-18, line 357-361).

6) Results obtained with the activator and dual TRs are not fully convincing. In particular, control EMSA assays have been provided only for the dual TR FleQ (Figure S3). EMSA should be provided also for ExsA, GacA, ErdR, AmrZ. In addition, transcriptional fusion experiments or RT-PCR experiments should be carried out to validate in vivo the EMSA results. Concerning FleQ, only half of the tested promoters showed a clear band-shift after binding of FleQ (PA2393, PA2619, PA2653, PA2955, PA4981). I understand that this is a lot of work, perhaps authors could limit their work to FleQ and another transcriptional activator.

Reply: Thanks for the critical comment. We agree with the reviewer that AIDmut-seq also identifies some false-positive results, as exemplified by several newly found FleQ targets (*fdx1* and *fimW*). And in vivo RT-PCR experiments should be conducted. In the revised manuscript, we focused on the two transcriptional regulators FleQ and ErdR in a separate part of the results section. The SNP spectrums from AIDmut-seq, the in vitro EMSA results, and the in vivo real-time RT-PCR data for FleQ and ErdR targeted promoters were collected together in a new figure (Figure 5).

Figure 5. Application of the AIDmut-Seq method to FleQ and ErdR in *P. aeruginosa*. (A) SNP spectrum obtained from the AIDmut-Seq result for FleQ (left), and the validation of FleQ binding to newly identified promoters using EMSA (right). Calculated TF-binding DNA motifs based on the AIDmut-Seq data are presented at the top right of each SNP spectrum. The upper motif was generated according to AIDmut-Seq data, lower motif represents the previously established PWM model. The number in parentheses after motifs indicates the number of promoters used to calculate the motif. Promoters known to be regulated by FleQ were annotated in the mutation spectra. (B) Quantification of the transcription of FleQ-targeted promoters in the wild type PAO1 and the *fleQ* mutant strains. (C) Quantification of the transcription of ErdR-targeted promoters in the wild type PAO1 and the *erdR* mutant strains. Statistical analysis was carried out by two-sample t-test. ns, non-significant; *, $p < 0.05$; **, $p < 0.01$; ***, $p < 0.001$. (D) SNP spectrum obtained from the AIDmut-Seq result for ErdR (left), and the validation of ErdR binding to newly identified promoters using EMSA (right).

Minor comments (partial revision, only up to line 137).

Reply: We thank the reviewer for these kind minor comments, we have revised the sentences according to the reviewer's suggestions.

Line 64, please explain Ugi gene function and why this is important; Lines 64-68- not very clear, please reformulate;

Reply: Sorry, we made a mistake. We want to say that 3D-Seq needs the excess expression of DddA cognate immunity determinant, not Ugi. And we have rewritten the sentences (64-68) in the revised manuscript as below (page 6, line 103-108).

“Besides, DddA targets double-stranded DNA and initiates multiple SNPs within a broad range of 10 kb, which is lethal to bacterial cells when essential genes were inactivated by mutation. Therefore, 3D-seq requires the exogenous expression of a DddA cognate immunity determinant induced by arabinose to antagonize DddA’s activity, so that genomic mutations only occurs when arabinose was removed.”

Line 71, AID from which organism?

Reply: Thanks, AID is from mouse. we have added the explanation in the revised manuscript (page 6, line 116).

Line 73, what do you mean for toxic? This is not clear enough, please reformulate, explain better;

Reply: Thanks, we have changed the misleading word “toxic” and reformulated the sentence in the revised manuscript (page 7, line 118-120), as below:

“Compared to DddA whose deaminase activity does not depend on the accession of single-stranded DNA, AID should produce fewer genomic mutations, and therefore be less harmful to cells.”

Lines 75-76, What is exactly the "dCas9-guided MS2-AID" mutation generator? It is likely that many readers could not know what the authors are talking about;

Reply: Thanks, we have added explanations in the revised manuscript (page 7, line 122-124) as below:

“As exemplified by the dCas9-guided MS2-AID mutation generator, in which AID was fused to a hairpin-binding protein and was recruited by dCas9 to generate mutations at target genomic sites”.

Line 76, please use instead of "will" use "could";

Reply: Thanks, we have changed the word according to the reviewer’s suggestion.

Line 85, please use "could cost" instead of "costs";

Reply: Thanks, we have revised the sentence according to the reviewer’s suggestion.

Line 88, what is exactly AIDΔ? What is the difference compared to the wild type AID?

Reply: Thanks, we have added explanations in the revised manuscript (page 7-8, line 137-139).

“AIDΔ is a variant of the mouse cytidine deaminase AID, with its nuclear export signal deleted and has three amino acid substitutions (K10E, E156G, T82I)”.

Line 95, The LasR inducer is named N-3-oxo-dodecanoyl-homoserine lactone and should be abbreviated as 3OC12-HSL;

Reply: Thanks, we have revised the inducer name throughout the manuscript.

Line 105, please add reference for the tested promoters;

Reply: Thanks, we have added references for the tested promoters in the revised manuscript.

Line 137, not very clear, please clarify. Perhaps authors want to say that "high expression levels of AID732-TF may lead to the detection of a large number of binding sites with weak binding

affinity".

Reply: Thanks, we have revised the sentence in the revised manuscript (page 10, line 192-195).

As below:

"Too low expression level of the fusion protein will result in failure in detecting expected binding sites, while too high expression level of AID732-TF may lead to the detection of a large number of binding sites with weak binding affinity."

Reviewer #2 (Comments for the Author):

In this manuscript, the authors developed a new method, called AIDmut-seq, to identify binding sites of transcriptional factor *in vivo*. The AIDmut-seq method is performed by three steps composing with fusion protein, extraction of genomic DNA and sequencing/SNP profiling. This approach is easy to be employed by junior and interdisciplinary investigators with only basic understanding in molecular biology. To establish the AIDmut-seq platform, the authors optimized the fusion direction, linker type, AID variant, induction time and arabinose concentrations. After which, the AIDmut-seq was conducted to examine binding sites of quorum sensing regulator LasR. Sequencing depths and reproducibility of SNPs in independent experiments were compared to evaluate the repeatability AIDmut-seq. Further, AIDmut-seq was applied to other transcriptional factors in *P. aeruginosa* such as FleQ, AmrZ, ExsA and etc.. Subsequently, EMSA was conducted to validate detected target sequences. Finally, the authors compared AIDmut-seq method with the classic Chip-seq method, and showed that AIDmut-seq has several advantages compared to Chip-seq. In summary, this study developed a useful approach for detecting protein-DNA binding sites which might support research field about transcriptional regulatory network.

I think this manuscript should be properly revised before its acceptance.

Reply: We thank the reviewer for the positive comment of our work. Special thanks to the reviewer's careful comments and advices, which improved the quality of our manuscript. The comments were addressed point by point.

Minor comments:

1) Line 20 and 35, "*in vivo*" should be written in italic.

Reply: Thanks, we have changed the format of "*in vivo*" and "*in vitro*" to italic throughout manuscript.

2) Line 64, SNPs full name should be shown at first time.

Reply: Thanks, we have added the full name of SNPs (single nucleotide polymorphisms) in the revised manuscript.

3) Fig. 3A-3L should be reordered according to the order they appear in the text, as well as Fig. 4A-4C.

Reply: Thanks, we have reordered the panels in Fig. 3 and other figures in the revised manuscript.

4) Manufacturer and affiliated states of reagents and kits in Methods should be provided.

Reply: Thanks, we have added the manufacturer's information for reagents and kits in the revised manuscript.

5) Line 359, "*Pseudomonas*" should be written in italic.

Reply: Thanks, we have revised the format in the text.

6) A large number of species names are not italicized, such as *Pseudomonas aeruginosa* at line 491 in References. The format of some references is wrong. For example, the first letter of each word of paper title are capitalization. The authors need revise.

Reply: Thanks, we have revised the reference format in the text.

7) Strains, plasmids and primers used in this study are listed in supplemental material.

Reply: Thanks, we have added the information of strains, plasmids, and primers used in the study in the revised supplementary material.

8) Line 25-26 and line 40 in supplemental material, "*P. aeruginosa*" and "*Escherichia Coli*" are revised in italic.

Reply: Thanks, we have revised the formats in the revised manuscript and supplementary material.

9) Line 40 in supplemental material, pET28a instead of pet28a.

Reply: Thanks, we have revised the format in the revised supplementary material.

10) Line 172 in supplemental material, LasR instead of lasR.

Reply: Thanks, we have revised the formats in the revised supplementary material.

December 1, 2022

Dr. Fan Jin
Chinese Academy of Sciences Shenzhen Institutes of Advanced Technology
Shenzhen
China

Re: Spectrum03783-22R1 (AIDmut-seq: A three-step method for detecting protein-DNA binding specificity)

Dear Dr. Fan Jin:

Your manuscript has been accepted, and I am forwarding it to the ASM Journals Department for publication. You will be notified when your proofs are ready to be viewed.

Sincerely,

Beile Gao
Editor, Microbiology Spectrum

Journals Department
Supplemental file 5: Accept
Supplemental file 3: Accept
Supplemental file 4: Accept
Supplemental file 1: Accept
Supplemental file 2: Accept